# Lymphatic endothelial-cell expressed ACKR3 is dispensable for postnatal lymphangiogenesis and lymphatic drainage function in mice

**Elena C. Sigmund[1], Lilian Baur[1], Philipp Schineis[1], Jorge Arasa[1], Victor Collado-Diaz[1], Martina Vranova[1], Rolf A. K. Stahl[2], Marcus Thelen[3], Cornelia Halin[1]\***

**1** Institute of Pharmaceutical Sciences, ETH Zurich, Zurich, Switzerland, **2** Universitätsklinikum, University of Hamburg, Hamburg, Germany, **3** Faculty of Biomedical Sciences, Institute for Research in Biomedicine (IRB), Università della Svizzera italiana, Bellinzona, Switzerland

\* cornelia.halin@pharma.ethz.ch

**Data Availability Statement:** All relevant data are accessible. Most relevant data are within the manuscript and its Supporting information files.

## Abstract

Atypical chemokine receptor ACKR3 (formerly CXCR7) is a scavenging receptor that has recently been implicated in murine lymphatic development. Specifically, ACKR3-deficiency was shown to result in lymphatic hyperplasia and lymphedema, in addition to cardiac hyperplasia and cardiac valve defects leading to embryonic lethality. The lymphatic phenotype was attributed to a lymphatic endothelial cell (LEC)-intrinsic scavenging function of ACKR3 for the vascular peptide hormone adrenomedullin (AM), which is also important during postnatal lymphangiogenesis. In this study, we investigated the expression of ACKR3 in the lymphatic vasculature of adult mice and its function in postnatal lymphatic development and function. We show that ACKR3 is widely expressed in mature lymphatics and that it exerts chemokine-scavenging activity in cultured murine skin-derived LECs. To investigate the role of LEC-expressed ACKR3 in postnatal lymphangiogenesis and function during adulthood, we generated and validated a lymphatic-specific, inducible ACKR3 knockout mouse. Surprisingly, in contrast to the reported involvement of ACKR3 in lymphatic development, our analyses revealed no contribution of LEC-expressed ACKR3 to postnatal lymphangiogenesis, lymphatic morphology and drainage function.

## Introduction

Atypical chemokine receptors (ACKRs) are chemokine receptors that do not signal via G proteins but instead act as scavenging receptors. ACKR3 binds and scavenges the chemokine ligands CXCL12 and CXCL11 [1,2]. In the case of CXCL12, ACKR3 is well known to shape local and systemic CXCL12 levels, thereby impacting its activity on CXCR4 [3–6]. By this means, ACKR3 exerts important functions during embryonic development, in leukocyte migration and in the regulation of the humoral immune response [7–11]. Furthermore, ACKR3 is upregulated in various types of cancer [12] and was found to be involved in tumor

Additional raw data for image analyses and lymphatic drainage experiments are deposited in the ETH Research Collection (DOI: 10.3929/ethz-b-000475154).

**Funding:** C.H. and M.T. gratefully acknowledge common financial support from the Swiss National Fund Sinergia program (CRSII3_160719 / 1). C.H. is supported by the ETH Zurich. The funders had no role in study design, data collection and analysis, decision to publish, or preparation of the manuscript.

**Competing interests:** The authors have declared that no competing interests exist.

cell migration [13], rendering it a potential therapeutic target for cancer therapies [13–16]. Besides CXCL11 and CXCL12, ACKR3 reportedly has non-chemokine ligands, namely, the vascular peptide hormone adrenomedullin (AM) [7], the intermediate opioid peptide Bam22 [17], as well as several other opioid peptides [18]. AM is a well-known inducer of endothelial cell proliferation [19], and influences blood vascular tone [20–23] and vascular permeability [24,25]. Besides its effects on blood vessels, AM also regulates cellular proliferation [19,26,27] and impacts permeability and drainage function in lymphatic vessels [20,26,28]. The scavenging activity of ACKR3 for AM was recently described in the context of a study investigating the role of ACKR3 in lymphatic development. Specifically, ACKR3-deficient mouse embryos were found to exhibit lymphatic hyperplasia and lymphedema [7], in addition to the previously described cardiac hyperplasia and valve defects [7,9,29,30]. The *in vitro* and *in vivo* findings presented in this study indicated that ACKR3 expressed by LECs acted as a scavenger of AM [7]. Consequently, loss of ACKR3 resulted in an overabundance of AM, leading to overshooting responses of LECs towards AM, what seemed to explain the lymphatic hyperproliferation and lymphedema phenotype observed in ACKR3-deficient mouse embryos [7,31].

To date, the role of ACKR3 in postnatal lymphatic vessels has only been marginally studied, likely because of the perinatal lethality of ACKR3-deficient animals [9,29,30] has impeded a thorough investigation of ACKR3's function during adulthood. Based on the reported findings on the lymphatic phenotype of ACKR3$^{-/-}$ mice during embryonic development and the fact that ACKR3 expression has already been documented in LECs in the kidney [32] and in cell culture [7,33], we hypothesized that ACKR3 expression in lymphatic vessels might be important for lymphatic vessel morphology and function throughout postnatal life. In this study, we therefore set out to investigate the expression and function of ACKR3 in the postnatal murine lymphatic vasculature. Using an *ACKR3$^{GFP}$* reporter mouse we found that ACKR3 was frequently expressed by LECs of lymphatic vessels in different tissues and in adjacent stromal cells and in blood vascular endothelial cells (BECs). To study the impact of ACKR3 deficiency during adulthood and circumvent the postnatal lethality of global ACKR3$^{-/-}$ mice, we further generated and validated a murine, tamoxifen- inducible lymphatic-specific ACKR3 knockout mouse model. Surprisingly, our results showed that—in contrast to the published role of ACKR3 in lymphatic development [7]—mature lymphatic vessel morphology and drainage function remained largely unaffected by postnatal ACKR3-deletion on lymphatic vessels.

## Material and methods

### Animals

*ACKR3$^{iΔLEC}$* mice were generated by intercrossing *C57BL/6 Prox-1-Cre$^{ERT2}$* mice [34] with *ACKR3$^{fl/fl}$Redstop$^{fl/fl}$* mice. *ACKR3$^{fl/fl}$* mice contain loxP-sites flanking *ackr3* and a Cre-inducible tandem-dimer red fluorescent protein reporter gene in the *ROSA26* locus (floxSTOPflox-RFP) [35] and were described previously [10]. Cre$^{ERT2}$-mediated *ackr3* deletion was induced in pups on p1-p3 by daily intragastric injection of 50μl of 1mg/ml tamoxifen (T5648, Sigma-Aldrich) dissolved in sunflower seed oil. *ACKR3$^{GFP/+}$* mice comprise a GFP reporter knocked into one copy of the *ackr3* gene (*C57BL/6-Ackr3tm1Litt/J*, the Jackson Laboratory) [8]. All mice were housed under opportunistic or specific pathogen free conditions at the HCI facility of the ETHZ Phenomics center. Adult animals were analyzed between 6–14 weeks while postnatal lymphangiogenesis was analyzed on p5 after birth. Adult animals were sacrificed using an overdose of anesthesia (160 mg kg−1 ketamine; 0.4 mg kg−1 medetomidine) followed by cervical dislocation, p5 pups by decapitation. All experiments involving mice were approved by the Cantonal Veterinary Office Zurich and performed according to the animal protocols ZH238/2017, ZH025/2017, 237/16 and ZH268/2014.

## FACS sorting and qRT-PCR

Ears from two animals/sample were pooled, cut into small pieces and enzymatically digested in a 4 ml mixture of 10 mg/ml collagenase IV (Thermo Fisher Scientific), 5 mg/ml dispase II (Sigma-Aldrich), 0.1 mg/ml DNase I (Roche) and 1 mM $CaCl_2$ (Sigma-Aldrich) in PBS (Thermo Fisher Scientific) for 15 min at 37˚C in rotation. The tissue suspension was minced through a 40 μm cell strainer (BD Biosciences), and the resulting single-cell suspensions were stained for 15 mins at 4˚C with APC/Cy7 anti-mouse CD45 (BioLegend), PE anti-mouse CD31 (BioLegend), APC anti-mouse Podoplanin (BioLegend), AF488 anti-mouse LYVE-1 (eBioscience) and 7-AAD (BioLegend). Using a FACS Aria Cell Sorter (70 μm nozzle) viable ECs were sorted into PicoPure RNA extraction buffer (Thermo Fisher Scientific) using the outlined sorting strategy. Subsequently, RNA was isolated and genomic DNA was eliminated using the PicoPure RNA isolation kit (Thermo Fisher Scientific). CDNA was prepared using Ovation Pico WTA System V2 (NuGen) and qRT-PCR was performed on a QuantStudio 7 Flex Real-Time PCR System (Applied Biosystems) using the SYBR Power Up™ SYBR Green Mastermix (Life Technologies) and following primer sequences: **mRPLP0 (FW)**: 5'-AGATTCGGGATATGCTGTTGGC-3', **mRPLP0 (RV)**: 5'-TCGGGTCCTAGACCAGTCTTC-3', **mACKR3 (FW)**: 5'-GAGGTCACTTGGTCGCTCTC-3', **mACKR3 (RV)**: 5'-GTGTCCACCACAATGCAGTC-3', **mAdm (FW)**: 5'- CTACCGCCAGAGCATGAACC-3', **mAdm (RV)**: 5'-GAAATGTGCAGGTCCCGAA-3', **mCXCL12_(FW)**: 5'-GGAGGATAGATGTGCTCTGGAAC-3', **mCXCL12 (RV)**: 5'-AGTGAGGATGGAGACCGTGGTG-3', **mCALCRL (FW)**: 5'-CAAGATCATGACGGCTCAATA-3', **mCALCRL (RV)**: 5'-CGTCATTCCAGCATAGCCAT-3' **mRAMP2 (FW)**: 5'-ACGAAACACATGTCCTACCTTGCTG-3',**mRAMP2** (**RV**): 5'- TCGCAAAGTGTATCAGGTGAGCCT-3', **mRAMP3 (FW)**: 5'-GGT CAT TAG GAG CCA CGT GT-3', **mRAMP3 (RV)**: 5'-GGG CTA AAC AAG CCA CAG CT-3'. Relative quantifications of gene expression were performed using the comparative cycle threshold method (ΔCT) with rplp0 (ribosomal protein lateral stalk subunit P0) as the reference gene. The values represent average relative gene expression in sorted LECs normalized to sorted BECs.

## CXCL11/12-AF647 uptake in isolated dermal stromal cells

Murine primary dermal LECs were isolated from 6-12-week-old *ACKR3*[iΔLEC], *ACKR3*[WT] littermates or *ACKR3*[WT]*ROSA*[iRFP] control mice. Ears were cut at the base of the ear, transferred into 1% Penicillin/Streptomycin (P/S) (Gibco) for 30 mins, split and cut into small fragments. The fragments were incubated in 0.25mg/ml Liberase DH (Roche) and 0.1mg/ml DNase I (Roche), activated with $Ca^{2+}$ and $Mg^{2+}$ in RPMI for 1h at 37˚C, with gentle agitation. Digested fragments of each animal were passed through a 70μm cell strainer, centrifuged and subsequently seeded into 4 wells of a collagen type I- and fibronectin- (Advanced BioMatrix/Sigma-Aldrich) (10μg/ml, each) coated, 6 well plate in α-MEM supplemented with 1% L-glutamine 10% FBS, 1% P/S (all Gibco). Unattached cells were removed by washing with PBS, the next day, and the medium was replaced every second day. When cells reached confluency, the scavenging assay was performed by addition of 50nM CXCL11/12- Alexa Fluor 647 (CXCL11/12-AF647) [36] and 1μM CCX771 (ChemoCentryx) or vehicle control diluted in α-MEM starvation medium containing 2% FBS and 1% P/S. The uptake was performed for 1 h at 37˚C, or at 4˚C for binding controls. Afterwards, cells were washed once with PBS and then subjected to a short acidic wash (100mM NaCl, 50mM glycine (Fluka, Sigma-Aldrich), HCl, pH3). After 1 min, the acidic buffer was replaced with PBS and cells were detached with Accutase® (Sigma-Aldrich) for 3 min at 37˚C. Eventually, cells were harvested in cold FACS buffer (2% FBS, 2mM EDTA in PBS), centrifuged, and stained with anti-CD31 FITC, (clone MEC 13.3, BD Biosciences), and anti-Podoplanin Alexa Fluor BV421 (clone 8.1.1, Biolegend) in FACS

buffer for 15 min at 4˚C. Cells were washed once with FACS buffer and then acquired on a Cytoflex S apparatus (Beckman Coulter, Brea, CA, USA) using CytExpert software and analyzed with FlowJo software 10.4.0 (Treestar).

## Wholemount immunostaining

**Ear skin.** Whole mount immunostainings were performed as described previously [37]. Briefly, mice were sacrificed and ear halves were fixed for 2h in 2% paraformaldehyde (PFA)/PBS at 4˚C. Subsequently, the ears were washed with 0.3% Triton-X/PBS and blocked for 2h in "Immunomix", containing 0.3% bovine serum albumin (Sigma-Aldrich) and 5% normal donkey serum (Sigma-Aldrich) in 0.3% Triton-X/PBS. Antibody staining with primary antibodies diluted in Immunomix was performed overnight at 4˚C in the dark. The following primary antibodies were used in ear whole mounts: rat anti-mouse CD31 (BD Pharmingen), rabbit anti-mouse LYVE-1 (Angiobio), mouse anti-mouse αSMA eFluor660 (ebioscience) rabbit anti-tubulinβ3 (Biolegend), hamster anti-mouse Podoplanin (clone 8.3.3). The following day, the samples were washed with 0.3% Triton-X/PBS, and then incubated for 3 h with appropriate secondary antibodies conjugated to Alexa-Fluorophores (Invitrogen). Samples were washed for 2 h with 0.3% Triton-X/PBS and mounted in Mowiol (Vector Laboratories).

**Lymphatic flank collector.** To harvest the flank collectors, the animal was pinned down on a silica plate and opened with a median skin cut, and two additional cuts, starting from the median incision towards the axilla and groin. The flank collectors were carefully excised from the surrounding tissue, using fine scissors and a stereomicroscope, after injection of Evans blue into the inguinal lymph node (LN) to increase the visibility of the flank collector. The harvested flank collectors were pinned down into silica-coated wells and stained in analogy to the staining protocol of ear skin (described above). Following primary antibodies were used for whole mount stainings of the flank collector: rat anti-mouse CD31 (BD Pharmingen) and goat anti-mouse Prox-1 (R&D).

## Image analysis of the ear skin

Confocal images represent maximum intensity projections of Z-stacks acquired using Confocal z-stacks on an LSM 880 (Carl Zeiss) confocal microscope using the Zen Software 2.3 (Carl Zeiss, Version 13.0.0.518). Images were processed with the Image Analysis Software IMARIS (Oxford Instruments, UK, Version 7.6.5). The LYVE1$^{+}$ lymphatic vessel network in the ear of $ACKR3^{i\Delta LEC}$ and $ACKR3^{WT}$ littermates was analyzed using the Autotube software as described before [38]. Three to five images per ear were analyzed and the values averaged per mouse. The experimenter who analyzed the images was blinded.

## Acute TPA-induced inflammation of the ear

Mice were anesthetized with 2.5% isoflurane. The baseline ear thickness was measured using an ear caliper (Brütsch Rüegger) before 1µg 12-O-tetradecanoylphorbol 13-acetate (TPA/PMA, Sigma-Aldrich), dissolved in acetone (Sigma-Aldrich), was applied to each side of the ear. Ear thickness was measured again, after 24 h, before animals were used to assess lymphatic drainage in the ear.

## Lymphatic drainage

Lymphatic drainage was measured as described previously [39,40]. In brief, $ACKR3^{i\Delta LEC}$ and $ACKR3^{WT}$ animals were placed in an IVIS imaging system (Caliper Life Sciences) and 0.05 nmol/g IRdye800-PEG-20 were injected s.c. using a 30G insulin syringe (Terumo, Tokyo,

Japan) with the ears taped down flat at the rim of the ear. The fluorescence intensity in the ear skin ($\lambda$ex = 745 nm, $\lambda$em = 800 nm, exposure time 4 seconds, binning 2) was measured at 0, 1, 2, 4, 6 and 24h. For analysis, ROIs were drawn around the ears and the average fluorescence intensity was measured in each ROI using Living image 4.0 software (Caliper Life Sciences, Hopkington, USA). After subtraction of the fluorescence intensity of uninjected ears, used as blank measurement, the average fluorescence intensity in each ROI was normalized to the initial average fluorescence intensity at time 0. The normalized average fluorescence intensities were plotted against time in Excel. Data points were fit according to first- order kinetics and the corresponding half-life ($t_{1/2}$) was determined ($T_{1/2} = \ln_{2/k}$). Lymphatic drainage in TPA-induced acute inflammation was measured one day after TPA-application and after determination of an increase in ear thickness.

## Statistical data analysis

Statistical analysis was performed using Prism 8 (GraphPad Software, LaJolla, CA, USA). Normally distributed data comparing two groups were analyzed using Student's t-test (comparing two groups) or one-way ANOVA (comparing multiple groups). Data that could not be considered normally distributed or failed normality testing, using the D'Agostino-Pearson test, were analyzed with non-parametric tests: Wilcoxon matched pairs test, Mann-Whitney U-test (comparing two unpaired groups) or Kruskal-Wallis test (comparing multiple unpaired groups). Data are shown as mean± standard error of the mean (mean ±SEM). Differences were considered statistically significant when $p < 0.05$, ns: not significant.

# Results

## ACKR3 expression by lymphatic vessels, blood vessels and different stromal cells during adulthood

Afferent lymphatic vessels can be structurally and functionally divided into initial lymphatic capillaries and downstream located lymphatic collectors (Fig 1A) [41,42]. To investigate the expression of ACKR3 in both lymphatic vessel segments, we isolated dermal LECs derived from either capillaries or collectors by FACS sorting from the ear skin of adult mice. The two LEC populations were FACS-sorted based on their either high or absent expression of the capillary marker LYVE-1 [42,43] (i.e. CD31$^+$Podoplanin$^+$LYVE-$^{high}$ capillary LECs (Cap LECs) and CD31$^+$Podoplanin$^+$LYVE-1$^-$; collector LECs (Coll LECs): gating strategy in (Fig 1B). Notably, we omitted the LYVE-1 intermediate cells, which presumably represent LECs from pre-collecting vessels [44]. In addition, also CD31$^+$Podoplanin$^-$ LYVE-1$^-$ BECs were sorted (Fig 1B). QPCR analysis performed on cDNA derived from the sorted cell samples identified *ackr3* expression in all three endothelial cell populations, particularly in capillary LECs (Fig 1C). Also, the ACKR3-ligands *cxcl12* and AM (*adm*) were present in LECs derived from both lymphatic capillaries and collectors (Fig 1D and 1E). Interestingly, the expression level of *ackr3* in capillary-type and collector-type LECs appeared to inversely correlate with the expression of *cxcl12* and *adm*. In support of our working hypothesis, i.e. that ACKR3 might be important for regulating LEC responses to AM during adulthood, we also detected the expression of the conventional AM 1 receptor subunits [44], namely the *calcitonin receptor- like receptor* (*calcrl*) (Fig 1F) and the *receptor activity-modifying protein 2* (*ramp2*) in LECs (and BECs) (Fig 1G). By contrast, expression of RAMP3 (*ramp3*), which can form a second AM receptor (AM2) in association with CALCRL (44), was detected in BECs but not in LECs (Fig 1H).

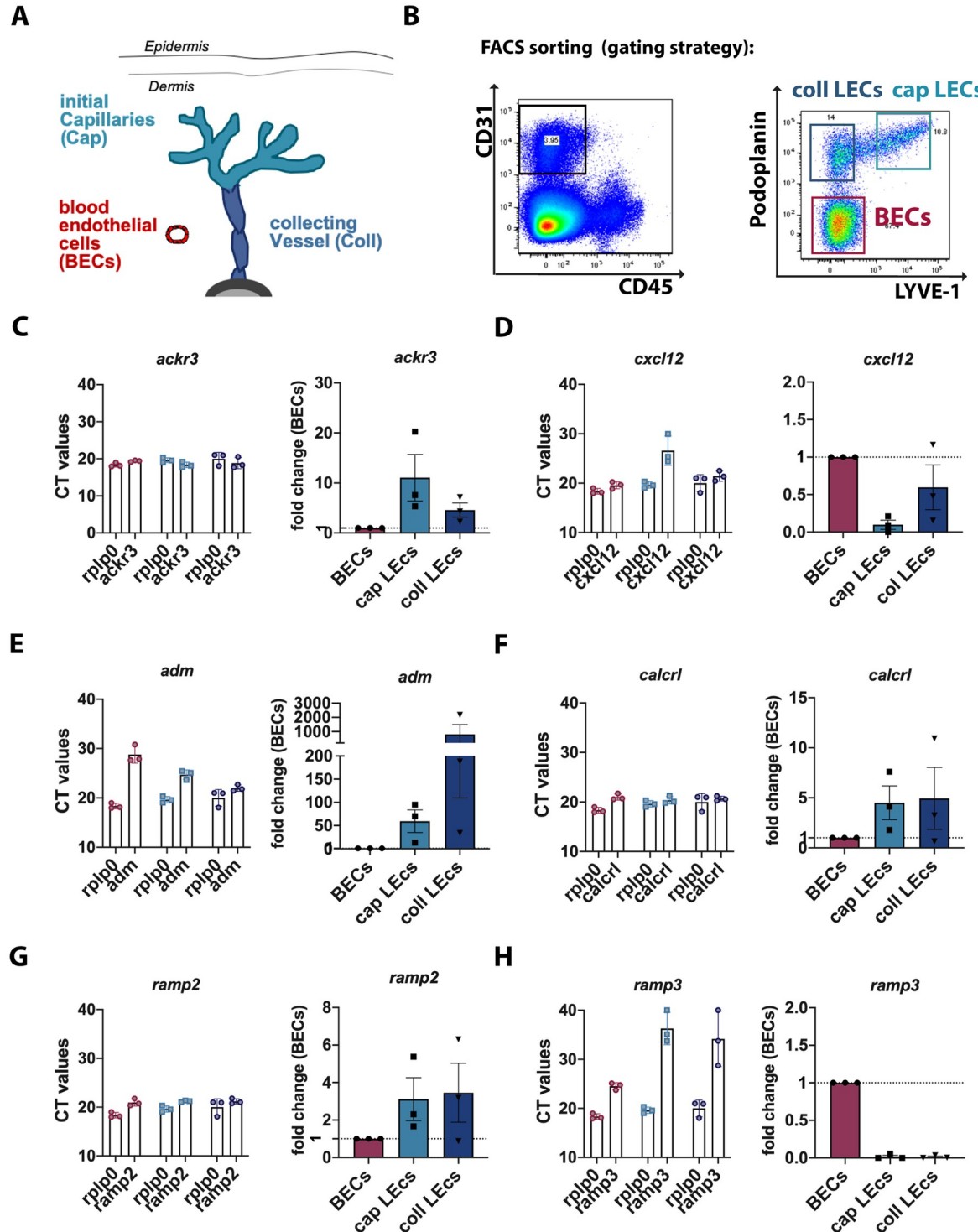

**Fig 1. Gene-expression signature of ACKR3, CXCL12, AM, CALCRL, RAMP2 and RAMP3 in BECs, collLECs and capLECs isolated from murine ear skin.** (A) Schematic localization of capLECs (cyan), collLECs (blue) and BECs (red) in afferent lymphatic vessels and blood vessels of the skin. (B) FACS sorting strategy: CD45$^-$ negative cells were sorted according to the following strategy: CD31$^+$Podoplanin$^-$ LYVE$^-$ blood endothelial cells (BECs), CD31$^+$Podoplanin$^+$ LYVE$^-$ collector LECs (collLECs) and CD31$^+$Podoplanin$^+$ LYVE$^+$ capillary LECs (capLECs). (C-H) qPCR analysis performed on cDNA derived from FACS-sorted capLECs, collLECs and BECs. Each Fig shows CT values compared to the housekeeping gene *rplp0* on the left, and the resulting relative mRNA expression of genes in capLECs and collLECs compared to BECs (set to one, dotted line), on the right. Results are shown for (C) *ackr3*, (D) *cxcl12*, (E) *adm*, (F) *calcrl* (G) *ramp2*, and (H) *ramp3*. Each data point represents the mean (mean ±SEM) of three independent qPCR replicates performed with cDNA of n = 3 samples, derived from 2 animals per sample.

Analysis of ACKR3 expression in murine tissues has been hampered by the lack of specific antibodies against murine ACKR3. In order to better characterize the expression pattern of ACKR3 within the lymphatic vasculature and surrounding tissues, we made use of adult *ACKR3*$^{GFP/+}$ reporter mice [8] and performed whole mount immunostainings in various tissues. In analogy to our qRT-PCR data (Fig 1C), GFP$^+$ cells were frequently found in both lymphatic capillaries and collecting vessels in the murine ear skin (Fig 2A–2C). The latter were identified by the expression of the lymphatic-specific markers Podoplanin or the transcription factor PROX-1 and by their characteristic morphology, including the presence of lymphatic valves (Fig 2A–2C). Notably, GFP was not exclusively expressed by LECs in *ACKR3*$^{GFP/+}$ reporter mice, but was frequently also found in unidentified or stromal cells (Fig 2A, 2C and 2D) in the surrounding tissue and associated with nerves positive for TUJ1 (Fig 2C and 2D). Moreover, GFP was also detected in many but not all Podoplanin$^-$/CD31$^+$ blood vessels (Fig 2B and 2C) and in the epidermal layer of the skin, indicating that ACKR3 is expressed by keratinocytes (Fig 2E). GFP expression was additionally detected in large lymphatic collectors such as the flank collectors of *ACKR3*$^{GFP/+}$ reporter mice (Fig 2F).

Since it was previously reported that mice with a lymphatic-specific, inducible deletion of CALCRL exhibited intestinal lymphangiectasia, causing protein-losing enteropathy and failure to recover from drug-induced inflammation [45], we were interested to see whether also ACKR3, is expressed by intestinal lacteals. Although GFP was strongly expressed in the blood vasculature of the small intestinal villi of *ACKR3*$^{GFP/+}$ reporter mice, we could not detect an overlap between the GFP signal and the lymphatic marker LYVE-1 expressed by LECs in lacteals (S1 Fig), indicating no expression of ACKR3 by lacteals. We further examined ACKR3 expression in large mesenteric lymphatic collectors and in the diaphragm shortly after birth at postnatal day (p) p5 (S2 and S3 Figs). In both of these tissues, lymphangiogenesis takes place postnatally or—in case of the mesenteric lymphatic collectors—is concluded postnatally [46–48]. In p5 mesenteries of *ACKR3*$^{GFP/+}$ reporter mice, we found GFP to be strongly expressed by large mesenteric blood vessels, specifically by veins. GFP expression was also detected in some lymphatic collectors, although the signal was comparably weaker (S2 Fig). In p5 diaphragms GFP expression was detected in lymphatic capillaries in some but not all areas of the tissue. Notably, GFP was also expressed by surrounding stromal cells (S3 Fig). Overall our findings demonstrate that ACKR3 is expressed in lymphatic vessels not only during murine embryonic development, as previously reported by Klein *et al.* [7], but also throughout adulthood.

## Generation and characterization of a lymphatic-specific, inducible ACKR3 knockout model

Depending on the genetic background, ACKR3$^{-/-}$ mice die during late embryonic development or postnatally due to severe defects of the cardiovascular system [9,29,30]. This lethality has thus far impeded studying the impact of ACKR3 deficiency on lymphatic vessels at the postnatal stage. To circumvent this problem, we generated a murine lymphatic-specific, tamoxifen-inducible ACKR3-knockout mouse line (*Prox1-Cre*$^{ERT2}$ x *ACKR3*$^{fl/fl}$ x *Redstop*$^{fl/fl}$; *ACKR3*$^{iΔLEC}$). *ACKR3*$^{iΔLEC}$ mice were obtained by crossing Prox1-Cre$^{ERT2}$ [34] animals with mice carrying *Ackr3*$^{flox}$ alleles [10] and a Cre- inducible tandem-dimer red fluorescent protein (RFP) reporter gene (floxSTOPflox-RFP) in the *ROSA26* locus [35] (Fig 3A). The latter allowed to asses Cre-induction efficiency by monitoring RFP expression. Gene deletion was induced by administration of tamoxifen to newborn pups on three consecutive days, starting on p1 after birth (Fig 3B). Efficient gene targeting was confirmed by almost uniform RFP expression in lymphatic vessels of the ear skin, detected in adult *ACKR3*$^{iΔLEC}$ mice (Fig 3C and 3D). In

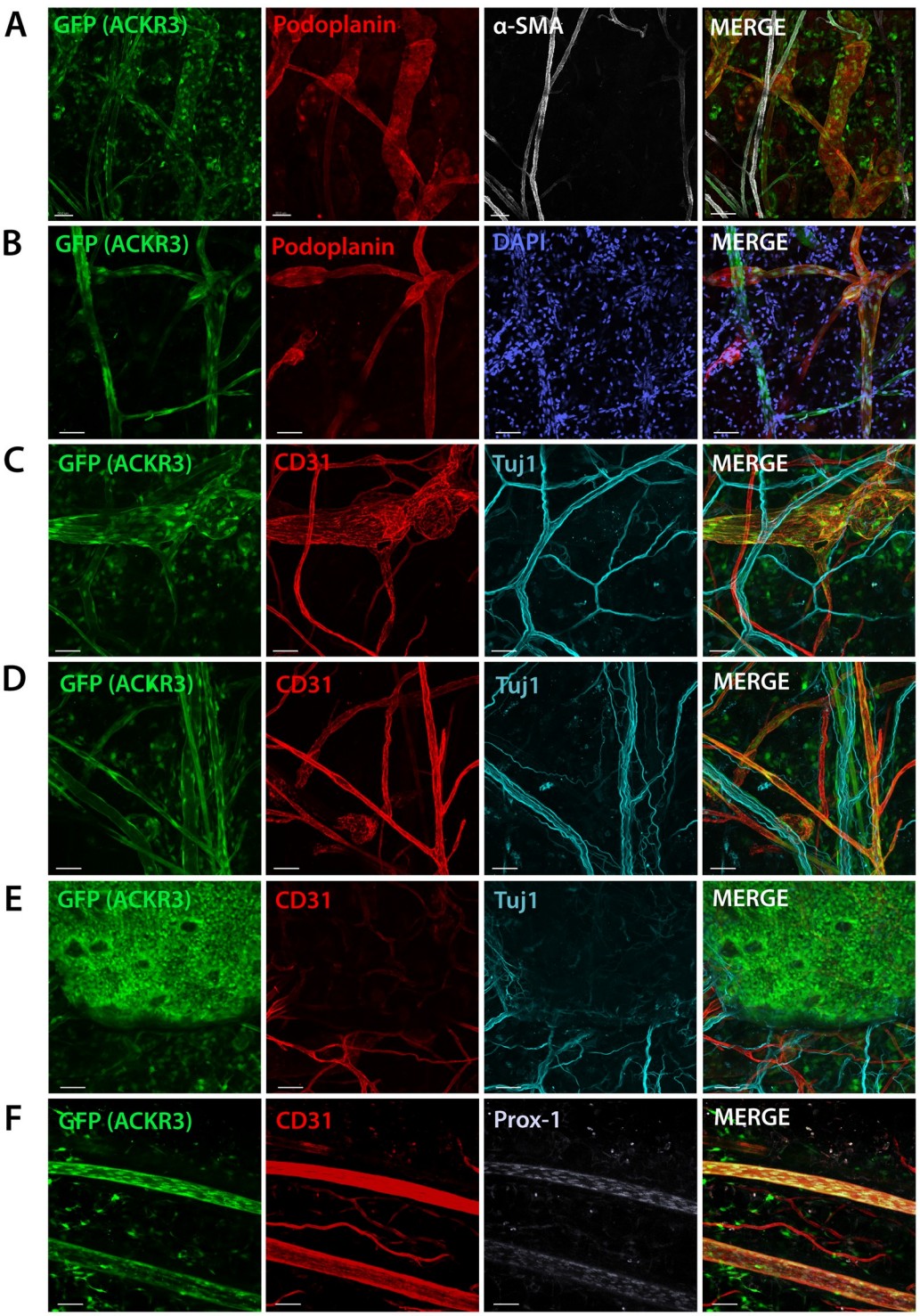

**Fig 2. Analysis of GFP expression in the ear skin and flank collector of adult *ACKR3*$^{GFP/+}$ reporter mice. (A-E)** Whole mounts were prepared from the ear skin of *ACKR3*$^{GFP/+}$ reporter mice. GFP was expressed by lymphatic vessels, blood vessels, various stromal cells and keratinocytes, in association with nerves, as evidenced by co-staining with **(A)** the lymphatic marker Podoplanin and alpha smooth muscle actin (αSMA), **(B)** Podoplanin and DAPI, **(C-E)** the panendothelial marker CD31$^+$ and the neuronal marker TUJ1. **(F)** Whole mount analysis performed on preparations of the flank collector confirmed GFP expression in large CD31$^+$ PROX-1$^+$ lymphatic collectors. Scale bar: 50μm Representative images of four separate experiments with n = 4 mice.

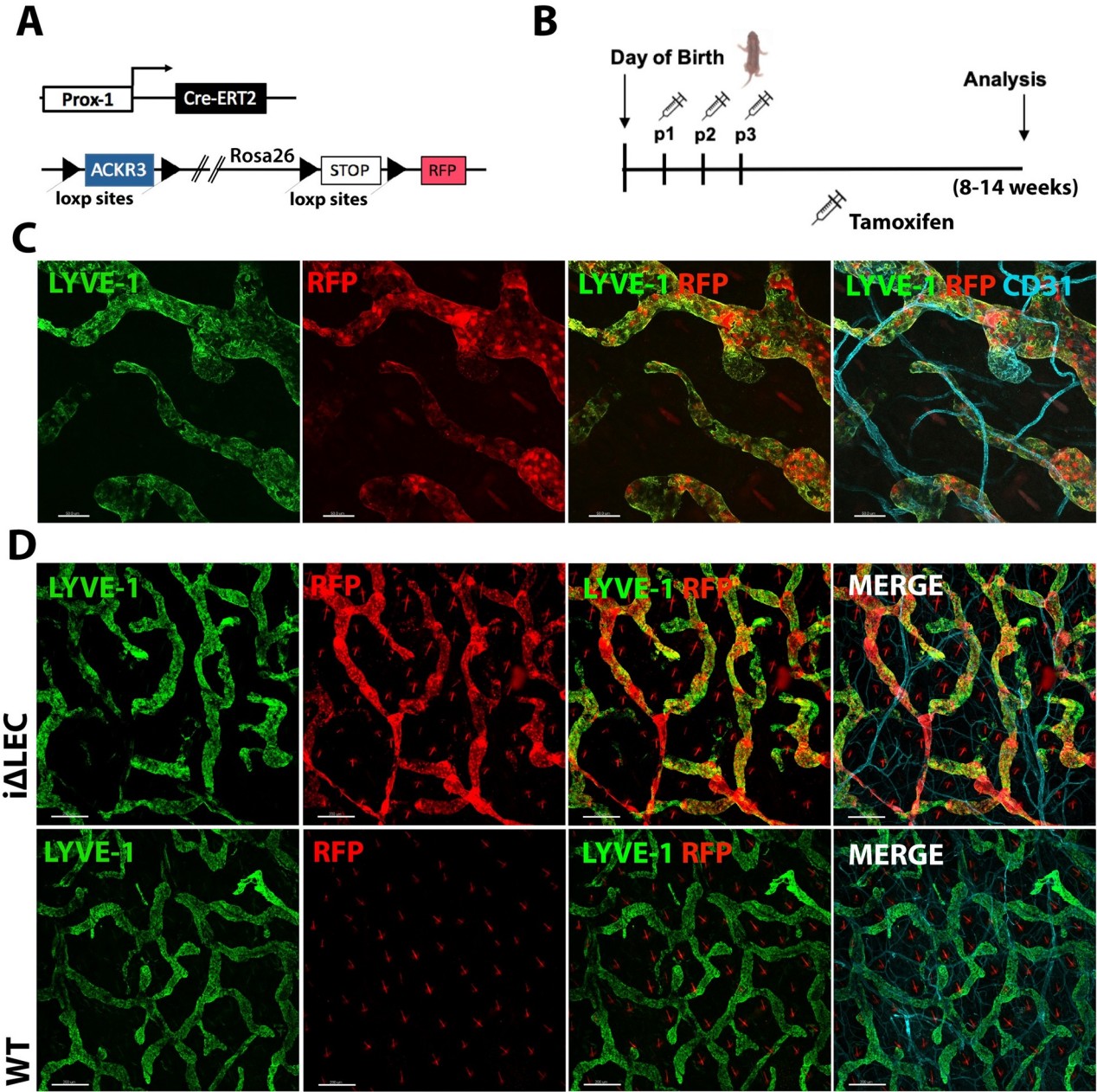

**Fig 3. Generation of a lymphatic-specific knockout mouse model (*ACKR3^{iΔLEC}*). (A)** *ACKR3^{iΔLEC}* animals express Cre^{ERT2} under the control of the lymphatic-specific Prox-1 promotor (*Prox-1^{CreERT2}*). *Ackr3 is* flanked by loxp-sites, in addition to an inducible RFP in the Rosa26 locus. **(B)** Tamoxifen was administered to newborn pups from p1-p3 by daily intragastric injections. **(C)** RFP induction in CD31⁺ LYVE-1⁺ lymphatic endothelial cells in the ear skin of adult *ACKR3^{iΔLEC}* mice. Scale bar: 100μm. (**D**) Overview: Almost uniform expression of RFP in lymphatic capillaries in *ACKR3^{iΔLEC}* mice. No RFP signal in *ACKR3^{WT}* littermate controls. Scale bar: 200μm. Representative images of three independent experiments and 12 animals per genotype analyzed.

agreement with this result, quantification of the RFP signal in LECs isolated from the ear skin by flow cytometry, indicated good penetrance and revealed that about 70–90% LECs were RFP⁺ (S4A and S4B Fig). ACKR3 knockdown was further confirmed by comparing *ackr3* mRNA expression in primary LECs isolated from the tail skin and from LNs of *ACKR3^{iΔLEC}* and *ACKR3^{WT}* mice by qPCR (S4C and S4D Fig). Based on the comparison of weight and

superficial appearance of $ACKR3^{i\Delta LEC}$ and $ACKR3^{WT}$ animals, we concluded that tamoxifen-treatment, itself, had no adverse effects on the general development of animals of either genotype (S4E Fig). Additionally, we compared the proliferative capacity of isolated primary tail LECs from $ACKR3^{i\Delta LEC}$ animals to those of $ACKR3^{WT}$ controls but found no difference in proliferation between ACKR3 deficient and sufficient tail LECs (S4F and S4G Fig).

In order to evaluate the efficiency of ACKR3 depletion in $ACKR3^{i\Delta LEC}$ animals at the protein level, we assessed the scavenging capacity of ACKR3 in primary dermal LECs isolated from $ACKR3^{i\Delta LEC}$ and $ACKR3^{WT}$ littermate controls. To assure that CXCL12 scavenging is exclusively mediated by ACKR3, we utilized an ACKR3-specific, recombinant, fluorescent chimeric chemokine, namely CXCL11/12-AlexaFluor647 (CXCL11/12-AF647) [36]. CXCL11/12 was shown to specifically bind to ACKR3, but—unlike CXCL12 and CXCL11 –does not bind to the conventional chemokine receptors CXCR4 and CXCR3. As a further control, we treated isolated dermal cells with the ACKR3-selective small molecule CCX771, which is a competitive agonist of ACKR3 and was previously shown to inhibit CXCL12 scavenging [13,49]. When incubating primary stromal cells isolated from the skin of either tamoxifen-treated $ACKR3^{i\Delta LEC}$ or $ACKR3^{WT}$ animals with CXCL11/12-AF647, we found that chemokine uptake was strongly reduced in LECs from $ACKR3^{i\Delta LEC}$ animals (Fig 4). Stromal cells and BECs, on the other hand exhibited CXCL11/12-AF647 uptake that was comparable to the wildtype control (Fig 4A). Although the extent of CXCL11/12-AF647 uptake was variable between different experiments (Fig 4B and 4C), the mean fluorescence intensity (MFI) detected in LECs from tamoxifen-induced $ACKR3^{i\Delta LEC}$ animals was consistently lower than in LECs from tamoxifen-induced WT animals and was comparable to the CCX771-treated control (Fig 4C). In summary, these results confirmed that in LECs from $ACKR3^{i\Delta LEC}$ mice, CXCL11/12-AF647 scavenging was abrogated, demonstrating that the applied tamoxifen regimen was effective in inducing ACKR3-deletion.

## ACKR3-deficiency does not affect postnatal lymphatic vessel development

Given that ACKR3 deficiency reportedly results in lymphatic hyperplasia during embryonic development [7], we investigated whether ACKR3 was equally important for the morphology and patterning of the lymphatic vascular network after birth. We chose three different organs/tissues, which are suitable to study postnatal lymphatic development. Namely, the ear skin, mesenteries and the diaphragm. Morphometric image analysis of the lymphatic network in the adult ear skin, which develops *de novo* after birth, did not reveal any significant difference in terms of LYVE-1$^+$ lymphatic vessel area, lymphatic vessel length, width or branching complexity between $ACKR3^{WT}$ and $ACKR3^{i\Delta LEC}$ animals (Fig 5A and 5B). In the diaphragmic muscle, lymphatic capillaries start to form between p0-p7, and most of the lymphatic development takes place postnatally [47]. Analysis of the LYVE-1$^+$ network was performed on whole-mount immunostainings of the diaphragm of $ACKR3^{i\Delta LEC}$ and $ACKR3^{WT}$ animals at p5 after birth, after tamoxifen treatment on p1-p3 (S5A Fig). At the time of analysis, LYVE-1$^+$ lymphatic structures in $ACKR3^{i\Delta LEC}$ mice uniformly expressed RFP, indicative of CRE-activity and ACKR3 deletion (S5B Fig). In line with the results from the ear, tamoxifen-induced ACKR3 deletion on p1-p3 did not impact postnatal lymphangiogenesis in the diaphragm (S5 Fig). Instead of an expansion of the LYVE-1$^+$ vessel area, as reported during embryonic development by Klein *et al.* [7], we observed a near-significant trend towards a decreased LYVE-1$^+$ area in $ACKR3^{i\Delta LEC}$ animals (S5C Fig), but no difference in vessel length, diameter, number of junctions or segments compared to $ACKR3^{WT}$ animals at p5 after birth (S5D–S5G Fig). Finally, we also analyzed the mesentery, in which lymphatic development starts around E13.5–14.5 but in which valve formation and vessel maturation continue until p8 after birth [46,48].

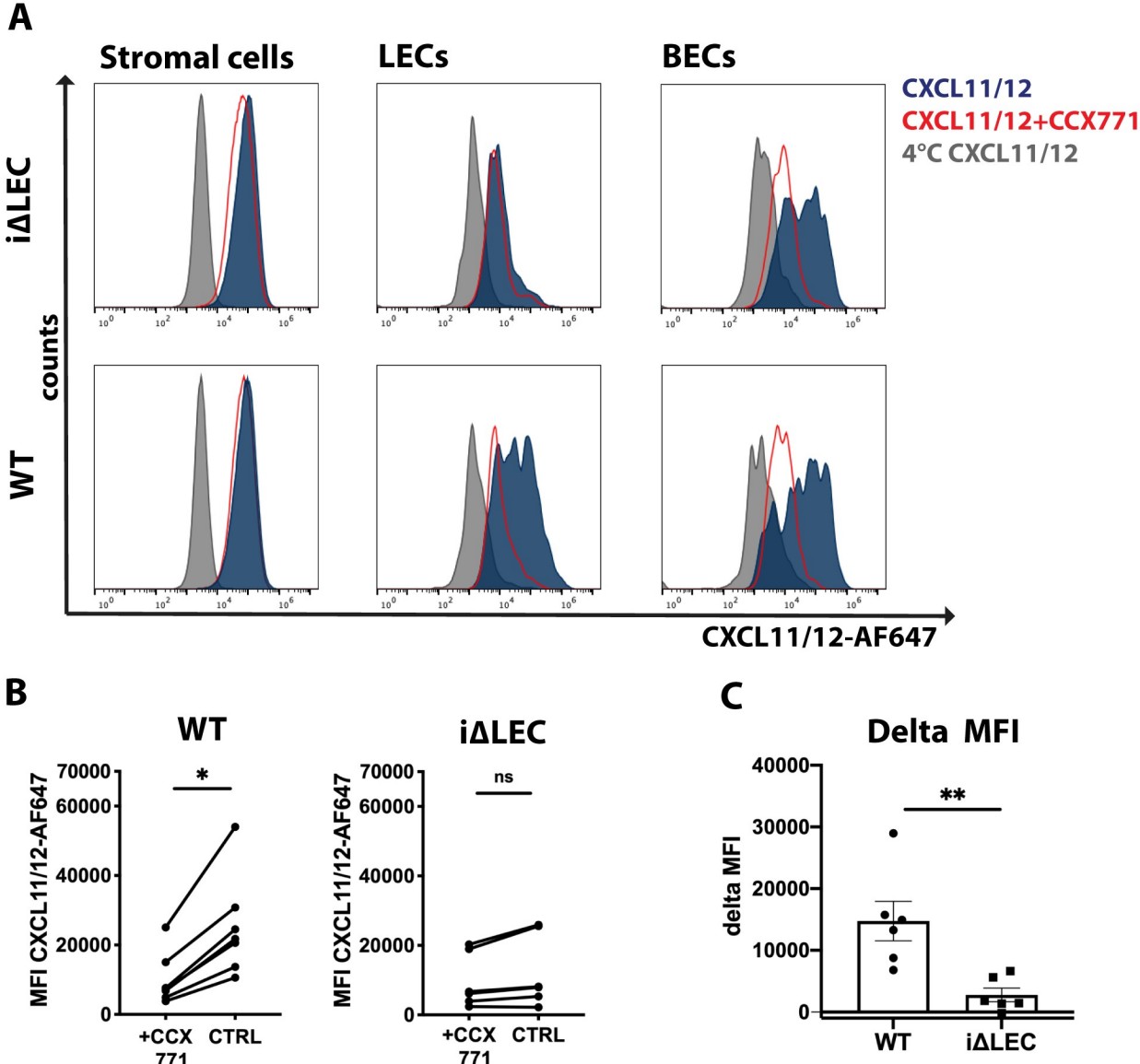

**Fig 4. CXCL11/12-AF647 scavenging is abrogated in dermal LECs isolated from *ACKR3^iΔLEC* animals.** Dermal stromal cells were isolated from the ear skin of *ACKR3^WT* or *ACKR3^iΔLEC* mice and their ability to scavenge the fluorescent ACKR3-specific chimeric chemokine CXCL11/12 was investigated *in vitro*. **(A)**. CXCL11/12-AF647 scavenging was specifically abrogated in dermal CD31⁺ Podoplanin⁺ LECs isolated from *ACKR3^iΔLEC* animals, or in LECs and CD31⁺ Podoplanin⁻ BECs from *ACKR3^WT* mice upon treatment with the ACKR3-selective competitive agonist CCX771. Representative histograms from 1 out of 5 experiments are shown. The corresponding gating strategy is shown in S4 Fig A. 4°C CXCL11/12: Chemokine uptake was performed at 4°C (impaired uptake control). CXCL11/12 + CCX771: Chemokine uptake was performed in presence of CCX771. **(B)** Comparison of the mean fluorescence intensity (MFI) values measured in chemokine uptake assays performed in presence (+CCX771) or absence (CTRL) of CCX771 in LECs isolated from either *ACKR3^WT* or *ACKR3^iΔLEC* mice, Wilcoxon matched-pairs test, n = 6 experiments. **(C)** Summary of the difference in MFI (ΔMFI) measured in the chemokine uptake assays performed in (B) in presence/absence of CCX771 in LECs isolated from either *ACKR3^WT* or *ACKR3^iΔLEC* mice. Mann-Whitney U-test, n = 6 experiments are shown in (B) and (C). Each dot represents the value obtained in one experiment, involving cells isolated from one animal.

Morphometric analysis of the mesenteric lymphatic vessels (S6A Fig) revealed no significant difference in the number of valves per vessel length, in the PROX-1⁺ vessel area, vessel length, vessel diameter, or number of segments per vessel length between *ACKR3^iΔLEC* and *ACKR3^WT* animals (S6B–S6F Fig). Together, these results demonstrated that the morphology of lymphatic vessels in adulthood is normal after postnatal deletion of ACKR3 in LECs.

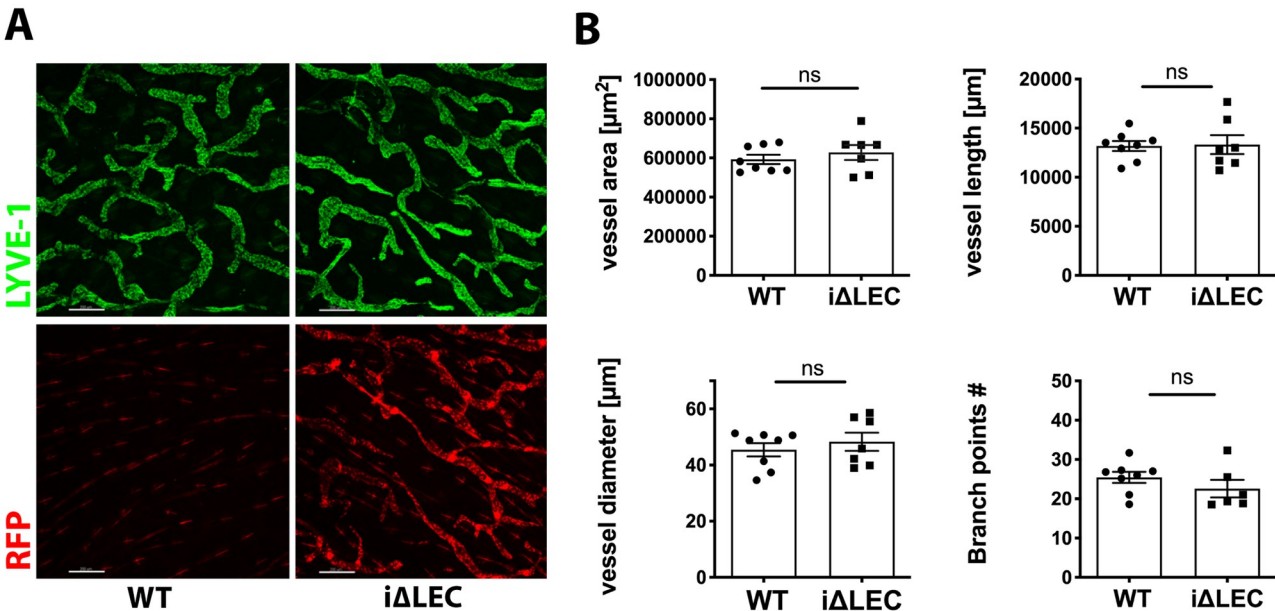

**Fig 5. Lymphatic vessel morphology and patterning was unaffected in the lymphatic network of the ear in adult *ACKR3^iΔLEC^* animals. (A)** Representative images of LYVE-1$^+$ capillary vessels network in the ear skin of adult *ACKR3^WT^* and *ACKR3^iΔLEC^*. **(B)** Image based-morphometric analysis of the LYVE-1$^+$ vessel area, the number of branch points, the vessel length, and vessel diameter showed no difference between *ACKR3^iΔLEC^* and *ACKR3^WT^* animals, unpaired Student's t-test, Pooled data from two independent experiments are shown and represented as mean ±SEM. Each data point corresponds to one mouse.

## Loss of ACKR3 does not impact lymphatic drainage

All of the components of the AM-signaling axis, namely, AM, CALCRL and RAMP2 have been shown to regulate blood vascular stability and permeability [24,25,50,51] and lymphatic drainage [20,26,45,52]. Since ACKR3$^{-/-}$ animals reportedly exhibit enlarged lymphatic vessels and edema during embryonic development (Klein et al., 2014), we sought to investigate whether ACKR3 is important for the permeability and drainage function of lymphatic vessels. To address this question, we performed a lymphatic drainage assay in the ear skin of adult *ACKR3^iΔLEC^* and *ACKR3^WT^* littermates. Specifically, an infrared 800 polyethylene glycol 20 (IRdye800-PEG-20) dye was injected intradermally into the ear skin and its drainage was measured by IVIS fluorescent imaging [39]. Comparing the half-life of the dye in uninflamed, steady-state ear skin, we observed no difference in drainage between *ACKR3^ΔiLEC^* and *ACKR3^WT^* littermate controls over the course of 24h (Fig 6A–6C). To exclude that a potential drainage defect might only manifest itself during tissue inflammation, i.e. a condition of enhanced vascular leakage and tissue edema, we further investigated lymphatic drainage in presence of skin inflammation induced by topical application of the skin irritant substance 12-O-tetradecanoylphorbol-13-acetate (TPA). 24h and 48h after TPA application the ears of *ACKR3^iΔLEC^* and control mice were similarly swollen (Fig 6D). Moreover, similarly as in uninflamed, steady-state ear skin, drainage was not impaired in *ACKR3^iΔLEC^* animals, as evidenced by a comparable tracer half-life found in both genotypes (Fig 6E and 6F). Thus, lymphatic-specific loss of ACKR3 did not result in any morphologic or functional differences that would be indicative of a lymphedema phenotype in adulthood.

## Discussion

By scavenging its chemokine ligands CXCL12 and CXCL11, ACKR3 exerts important functions in embryonic development, in the regulation of leukocyte and tumor cell migration

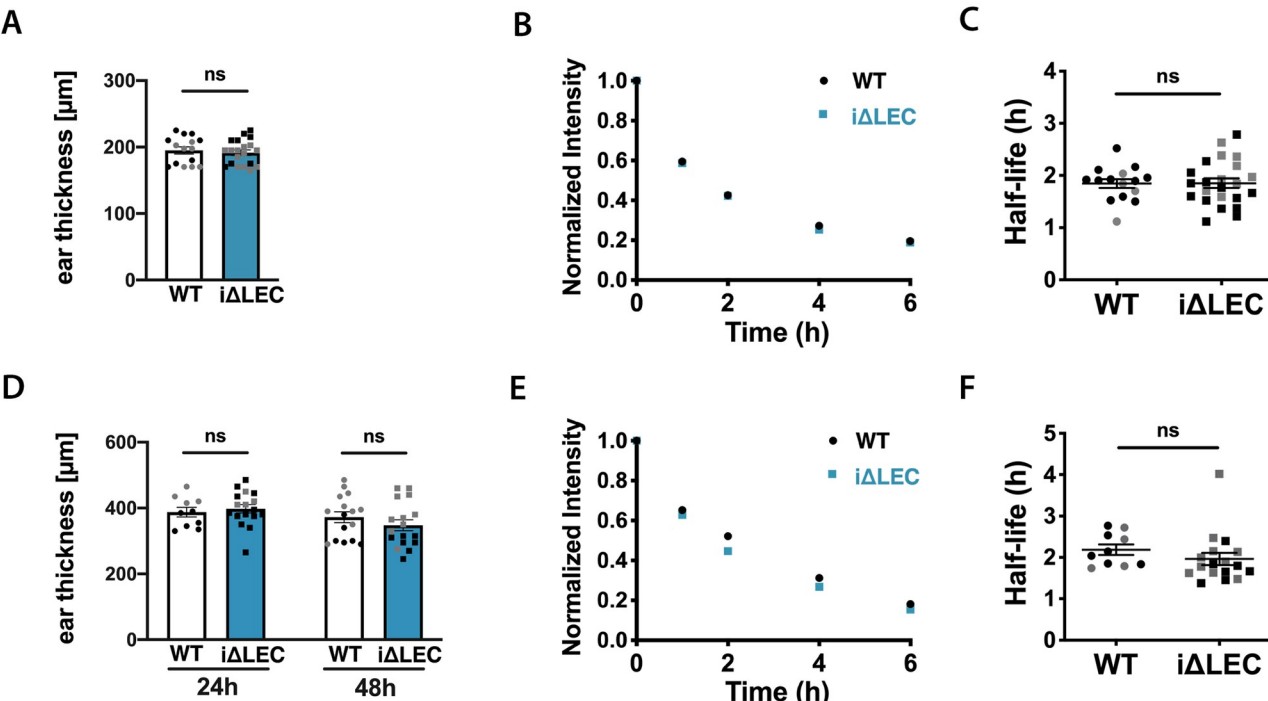

**Fig 6. Lymphatic drainage function was unaffected in *ACKR3*<sup>iΔLEC</sup> animals compared to *ACKR3*<sup>WT</sup> animals in steady-state.** (A) No difference in ear thickness in steady-state was measured between *ACKR3*<sup>iΔLEC</sup> and *ACKR3*<sup>WT</sup> animals. (B, C and E, F) Mice were injected with an IRdye800-PEG-20 dye *i.d.* in the ear skin. Clearance of the dye was monitored over 24h using an IVIS imaging system. (B) Average clearance plot of IRdye800-PEG-20 and (C) half-life in steady state. (D) Average ear thickness in males and females after induction of inflammation by topical application of TPA, at time point t = 24h and t = 48h. (E) Average clearance plot of IRdye800-PEG-20 and (F) half-life in TPA-induced acute inflammation. Pooled data from 3 independent experiments (or 2 independent experiments in TPA inflammation) are shown and represented as mean ±SEM. Mann-Whitney U- test, each dot represents one animal, females are depicted in black and males in grey color in dot plots.

across blood vessels and in B cell immunity/activation in germinal centers [8–11,13,16]. Recent findings have furthermore implicated ACKR3 in lymphatic vessel morphology and function, by identifying ACKR3 as a scavenging receptor for AM, which prevents overshooting AM responses in the lymphatic vasculature during development [8]. However, thus far, ACKR3 expression and function in postnatal lymphatic vessels have not been investigated, but could be important considering that ACKR3 is currently under investigation as a drug target in cancer therapy [12–15,53].

Our qPCR data of sorted dermal LECs and BECs indicated that ACKR3 is expressed in all endothelial cell types analyzed. Aside from the ligands of ACKR3, we also detected the expression of the AM1 receptor, consisting of CALCRL and the accessory protein RAMP2, in lymphatic capillary and collector type LECs. The detection of the conventional AM1 receptor genes in our study strengthened previous reports implicating AM in the direct regulation of vascular permeability of lymphatic vessels during adulthood [20,26,45,52]. Moreover, the confirmation of ACKR3, AM and AM receptor expression in LECs suggested that, in theory, ACKR3 might also be acting as scavenging receptor for AM and preventing overshooting AM activity in adult lymphatics [7]. Our whole mount stainings performed in *ACKR3*<sup>GFP/+</sup> reporter mice revealed frequent GFP expression in capillary- and collector-type lymphatic vessels in the ear skin. Notably, our whole mount images indicate that the level of ACKR3 expression in lymphatic vessels may vary between different tissues and organs. In particular, we found that skin lymphatic capillaries consistently expressed ACKR3, while GFP was detected

to a lesser degree in the diaphragm muscle and not at all in lacteals, although it was ubiquitously expressed in the surrounding blood vasculature of the villi. Accordingly, ACKR3 was frequently expressed in lymphatic collectors in the skin, while it was only weakly detected in mesenteric collectors. Currently, we do not know what drives these tissue-specific differences and what functional implications they might have. In addition, our fluorescent chemokine uptake studies showed that stromal cells isolated from the skin of WT mice were able to scavenge ACKR3-specific CXCL11/12-AF647, whereas its uptake into LECs derived from *ACKR3*[i∆LEC] mice was abrogated. This confirms that the dermal GFP+ cell types identified as ACKR3 expressers in the *ACKR3*[GFP/+] reporter mouse indeed expressed ACKR3 and indicates that the *ACKR3*[GFP/+] reporter mouse faithfully reports ACKR3 expression. Moreover, it shows that ACKR3 expressed in postnatal dermal lymphatics is functional.

Klein *et al.* have recently suggested that LEC-expressed ACKR3 serves to protect LECs from overshooting responses towards AM [7,31]. Global loss of ACKR3 resulted in lymphatic vessel hyperplasia and hyperproliferation in E13.5 embryos and an embryonic lymphedema phenotype [7]. Surprisingly, we found that the morphology of lymphatic vessels in the mature lymphatic network of the adult ear skin was unaffected by postnatal lymphatic-specific ablation of ACKR3. Lymphatic-specific ablation of ACKR3 was confirmed by qRT-PCR and detection of reduced relative *ackr3* expression in isolated primary LECs of ACKR3[i∆LEC] mice, compared to *ACKR3*[WT] controls, but more importantly by loss of protein function, evidenced by abrogated CXCL11/12- AF647 chemokine scavenging. Considering that the murine ear and its associated vasculature develops *de novo* after birth, this finding shows that LEC-expressed ACKR3 is not only dispensable for the maintenance of adult lymphatic vessel morphology, but also for postnatal development of lymphatic vessels in the ear skin. The fact that we did also not observe any lymphatic phenotype in the diaphragm at p5 after birth or in the expanding p5 mesenteric collectors further supports the notion that LEC-expressed ACKR3 does not regulate postnatal lymphangiogenesis. Similarly, we did not detect any difference in dermal lymphatic drainage between WT and *ACKR3*[i∆LEC] mice, neither under steady-state nor under inflammatory conditions.

One potential reason for the discrepancy between our results and the previously reported lymphatic phenotype [7] could be that LEC-specific deletion of ACKR3 does not phenocopy the lymphatic defects observed in a global ACKR3 knockout setting. As also evidenced by our whole-mount analyses in *ACKR3*[GFP/+] reporter mice, ACKR3 is not exclusively expressed in lymphatic vessels, but also present in keratinocytes in the skin and in BECs and stromal cells in most tissues, in addition to ACKR3's reported expression in cardiomyocytes [7,30,36]. Thus, even though the findings of Klein et al. suggested a cell-intrinsic AM scavenging function of ACKR3 in LECs [7,31], it is possible that lymphatic specific ablation of ACKR3, as investigated in our model, does not suffice to substantially modulate AM tissue levels and to induce the lymphatic phenotype observed in global ACKR3-deficient embryos. However, it needs to be kept in mind that, in addition to the AM1 receptor, dermal LECs also express AM ([7,26] and Fig 1). Thus, LEC-expressed ACKR3 would be expected to be most effective in fine-tuning and reducing AM levels in the dermal LEC micro-environment, rendering this explanation somewhat less likely.

A second potential explanation for the discrepancy between the lymphatic phenotype observed in ACKR3-deficient embryos [7] and the unaffected lymphatic vascular morphology and drainage function in adult *ACKR3*[i∆LEC] mice could lie in differences in AM concentrations in embryonic versus adult tissues. It is perceivable that a certain threshold concentration of AM needs to be reached before LEC proliferation, lymphatic permeability and lymphatic drainage are affected. Although not reported so far, it is possible that normal dermal tissue levels of AM in adult animals are lower than in embryos and thus fail to reach the threshold

concentrations necessary to impact lymphatic responses in adult $ACKR3^{i\Delta LEC}$ animals. However, it is technically challenging to assess potential differences of local AM concentrations in embryonic- compared to adult tissues experimentally: most methods that quantitatively assess protein levels in tissues rely on tissue lysates, where quantitative information about AM levels in the tissue microenvironment of LECs would be lost. In addition to differences between adult and embryonic AM tissue levels, it is perceivable that—besides ACKR3 expression—other regulatory mechanisms could render LECs in adult skin less responsive to AM. Finally, it is worth mentioning that another study recently reported that excess/high concentrations of AM could not displace fluorescent CXCL12 binding to ACKR3 overexpressed in U87 cells [54], thereby questioning the AM binding and scavenging functions of ACKR3. However, since this study was performed in a cell line, which does not naturally express ACKR3, and which might be lacking other co-factors necessary for AM binding, these findings do not necessarily allow to draw conclusions on physiological actions of ACKR3, such as AM scavenging in LECs.

Considering that we could not detect a contribution of LEC-expressed ACKR3 to postnatal lymphangiogenesis and lymphatic drainage, we suspect that the function of this receptor may lie more in the regulation of leukocyte trafficking. Of interest in this regard, CXCL12 reportedly is expressed by lymphatic vessels, and blockade of CXCR4 was previously shown to reduce migration of dendritic cells (DCs) from skin to draining lymph nodes [55]. However, our adoptive transfer studies performed with bone marrow-derived dendritic cells (BM-DCs), which are responsive for CXCL12 (S7A and S7B Fig), did not reveal any contribution of LEC expressed ACKR3 to DC migration to dLNs, neither in steady-state nor under inflammatory conditions (S7C–S7F Fig). It remains possible that LEC-expressed ACKR3 might still be required for DC migration from other tissues, or modulate migration of other CXCR4-expressing leukocytes e.g. neutrophils [56] or of cells expressing receptors for the other ACKR3 ligands, i.e. CXCL11. Taken together, our study shows that ACKR3 is frequently expressed in lymphatic vessels in various different tissues and acts as a functional scavenging receptor for its chemokine ligands in isolated and cultured primary LECs. At the same time, we could not detect any impact of ACKR3-deficiency in LECs on lymphatic morphology and drainage function during adulthood. Future studies will need to address, whether long-term systemic blockade of ACKR3 or global ACKR3 depletion in adult animals could result in a similar lymphatic phenotype as observed during lymphatic development.

## Supporting information

**S1 Fig. GFP is not expressed in intestinal lacteals or $ACKR3^{GFP/+}$ reporter mice. (A)** GFP was highly expressed in LYVE-1$^-$ CD31$^+$ blood vessels but not detected in LYVE-1$^+$ CD31$^+$ lacteals in villi of the small intestine. Scale bar: 100μm. (**B**) Higher magnification of a single villi in the duodenum. Scale bar: 30μm. Representative images of one experiment with n = 3 animals.
(TIF)

**S2 Fig. In the p5 mesentery of $ACKR3^{GFP/+}$ reporter mice, GFP is strongly expressed in veins and to a lower extent in lymphatic collectors. (A)** Overview picture of large blood vessels and PROX-1$^+$ lymphatic collectors. Scale bar: 200μm (**B**) Higher magnification shows GFP expression in lymphatic collectors. Scale bar: 100μm. Representative images of one experiment with n = 4 animals.
(TIF)

**S3 Fig. GFP is highly expressed by stromal cells in the diaphragm and occasionally by developing lymphatic capillaries of *ACKR3$^{GFP/+}$* reporter mice. (A)** Overview picture of the diaphragmic muscle in an area with high GFP expression in PROX-1$^+$ lymphatic vessels. Scale bar: 200μm **(B)** Higher magnification. Scale bar: 100μm. Representative images of one experiment with n = 4 animals.
(TIF)

**S4 Fig. Characterization of tamoxifen-induced ACKR3 depletion in *ACKR3$^{iΔLEC}$* mice. (A)** Gating strategy used for the detection of RFP$^+$ LECs in dermal stromal cell cultures isolated from ears of adult tamoxifen-treated animals. **(B)** % of RFP$^+$ LECs in comparison to the % of RFP$^+$ BECs and CD31$^-$ stromal cells, one-way ANOVA, n = 5 animals. **(C)** Relative *ackr3* mRNA expression in isolated dermal tail LECs, n = 4 animals. **(D)** Relative *ackr3* mRNA expression in isolated LN LECs, n = 3 animals. **(E)** Average body weight of 7-week-old female *ACKR3$^{iΔLEC}$* compared *to ACKR3$^{WT}$* control animals, Student's t-test, n = 8 animals. **(F)** Gating strategy for the detection of the proliferation marker Ki-67 in isolated dermal tail LECs. **(G)** Proliferation was unaffected in dermal tail skin LECs isolated from *ACKR3$^{iΔLEC}$* mice compared to *ACKR3$^{WT}$* controls, n = 4 experiments, each involving LECs isolated from one ACKR3$^{iΔLEC}$ and one ACKR3$^{WT}$ mouse. All data in (C- D, G) were analyzed using a Mann-Whitney U- test.
(TIF)

**S5 Fig. Postnatal deletion of ACKR3 did not cause a hyperplastic phenotype during postnatal lymphangiogenesis in the diaphragm. (A)** Lymphangiogenesis in the diaphragm was analyzed at p5 following treatment with tamoxifen at p1-p3 in two vessel segments per pup (red frames). **(B)** Robust RFP expression in lymphatics was detected in *ACKR3$^{iΔLEC}$* animals at p5. Scale bar: 300μm. **(C)** Image based-morphometric analysis of the LYVE-1$^+$ vessel area, **(D)** total vessel length in %, **(E)** vessel diameter, **(F)** number of junctions/branch points per vessel length and **(G)** the number of segments per vessel length showed no significant difference between *ACKR3$^{iΔLEC}$* and *ACKR3$^{WT}$* animals. Each data point derives from one pup (n = 9–11) and represents an averaged value of 2–3 quantified diaphragmic images. Student's t- test.
(TIF)

**S6 Fig. Postnatal deletion of ACKR3 did not affect the number of valves/vessel length in the mesentery lymphatic network. (A)** Analyzed area of the mesenteric lymphatic network and RFP reporter expression in PROX-1$^+$ lymphatic vessels at p5 after tamoxifen treatment at p1-p3. Scale bar: 200μm. **(B)** Comparison of the number of valves/vessel length between *ACKR3$^{iΔLEC}$* and *ACKR3$^{WT}$* animals. **(C)** Absolute PROX-1$^+$ area in [μm$^2$]. **(D)** Total lymphatic vessel length in [μm]. **(E)** vessel diameter in [μm], **(F)** Number of segments per total vessel length. Data from three independent experiments are shown as mean ±SEM. Each data point represents an averaged value of 3–5 quantified images of one pup. Student's t test.
(TIF)

**S7 Fig. Migration of adoptively transferred DCs to the popliteal LN was not impaired in *ACKR3$^{iΔLEC}$*mice. (A)** BM-DCs and LPS matured BM-DCs express CXCR4, n = 4 independent experiments, one-way ANOVA **(B)** CXCL12-mediates transmigration of BM-DCs through monolayers of immortalized LECs, n = 3 independent experiments. Kruskal-Wallis test. **(C)** CFSE$^+$ BM-DCs were adoptively transferred into steady-state and TPA-inflamed footpads of ACKR3$^{WT}$ and ACKR3$^{iΔLEC}$ animals. Draining popliteal LNs were harvested after 18 h and single cell suspension stained for CD45, CD11c, MHCII and analyzed by FACS. **(D)** Gating scheme. **(E, F)** The following quantifications in LN draining **(E)** steady-state or

(**F**) TPA-inflamed footpads are shown: Total numbers of CD45$^+$ cells and total number of migratory CFSE$^+$ and CFSE migratory DCs (CD45$^+$CD11c$^+$MHCII$^+$), percentage of migratory DC amongst CD45$^+$ cells and percentage of migrated, adoptively transferred DCs (CD45$^+$CD11c$^+$MHCII$^+$CFSE$^+$) amongst CD45$^+$ cells. Pooled data of three independent experiments, each dot represents the value measured in one animal, Mann- Whitney U-test. (TIF)

**S1 Methods. Supplementary experimental methods (related to S1–S7 Figs).**
(DOCX)

## Acknowledgments

The authors thank Taija Makinen (Uppsala University, Sweden) for kindly providing the *Prox1Cre^ERT2* mouse strain and Thomas Schall (ChemoCentryx Inc., Mountain View, CA, USA) for providing CCX771. Furthermore, the authors thank Angela Vallone, Ioannis Kritikos and the Scientific Center for Optical and Electron Microscopy (ScopeM) for excellent technical assistance, and the staff of the ETH Rodent Center HCI for animal husbandry. Moreover, we would like to thank Mona Friess for helpful discussions and her insights regarding specific techniques.

## Author Contributions

**Conceptualization:** Elena C. Sigmund, Cornelia Halin.

**Formal analysis:** Elena C. Sigmund, Lilian Baur.

**Funding acquisition:** Marcus Thelen, Cornelia Halin.

**Investigation:** Elena C. Sigmund, Lilian Baur, Philipp Schineis, Victor Collado-Diaz.

**Methodology:** Elena C. Sigmund, Philipp Schineis, Martina Vranova.

**Project administration:** Cornelia Halin.

**Resources:** Jorge Arasa, Rolf A. K. Stahl, Marcus Thelen.

**Supervision:** Cornelia Halin.

**Validation:** Jorge Arasa.

**Visualization:** Elena C. Sigmund, Lilian Baur.

**Writing – original draft:** Elena C. Sigmund, Cornelia Halin.

**Writing – review & editing:** Elena C. Sigmund, Marcus Thelen, Cornelia Halin.

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
