## [Decision Letter · Decision Letter 0]

4 Sep 2020

PONE-D-20-24551

Lymphatic endothelial-cell expressed ACKR3 is dispensable for postnatal lymphangiogenesis and lymphatic drainage function

PLOS ONE

Dear Dr. Halin,

Thank you for submitting your manuscript to PLOS ONE. After careful consideration, we feel that it has merit but does not fully meet PLOS ONE’s publication criteria as it currently stands. Therefore, we invite you to submit a revised version of the manuscript that fully addresses the points raised during the review process.

We look forward to receiving your revised manuscript.

Kind regards,

Jean-Léon Thomas

Academic Editor

PLOS ONE

2. To comply with PLOS ONE submissions requirements, in your Methods section, please provide additional information regarding the experiments involving animals and ensure you have included details on (1) methods of sacrifice and (2) the number of animals used in the experiment.

Additionally, please consider modifying your title to ensure that it is specific, descriptive, and concise (for example by specifying the animal model used)

Reviewers' comments:

Reviewer's Responses to Questions

**Comments to the Author**

1. Is the manuscript technically sound, and do the data support the conclusions?

Reviewer #1: Yes

Reviewer #2: Yes

2. Has the statistical analysis been performed appropriately and rigorously? 

Reviewer #1: Yes

Reviewer #2: No

3. Have the authors made all data underlying the findings in their manuscript fully available?

Reviewer #1: Yes

Reviewer #2: Yes

4. Is the manuscript presented in an intelligible fashion and written in standard English?

Reviewer #1: Yes

Reviewer #2: Yes

5. Review Comments to the Author

Reviewer #1: Sigmund et al. investigated the expression and function of the atypical chemokine receptor ACKR3 in the lymphatic vasculature of adult mice. The authors show that ACKR3 is widely expressed in mature lymphatics and that it exerts chemokine-scavenging activity in cultured murine skin-derived LECs. They generated and validated a lymphatic-specific, inducible ACKR3 knockout mouse that revealed no contribution of LEC-expressed ACKR3 to postnatal lymphangiogenesis, lymphatic morphology and drainage function.

This is the first study reporting ACKR3 expression and function in postnatal lymphatic vessels. Experiments and statistics are described in sufficient detail. Conclusions are presented in an appropriate fashion and are supported by the data. Limitations of the study are also discussed. However, the study would benefit from clarifying a few points:

1. Authors report that CXCL11/12-AF647 uptake is abrogated in primary LECs from tamoxifen-induced ACKR3i�LEC mice, demonstrating that the tamoxifen was effective in inducing ACKR3-deletion.

- What is the functional impact of ACKR3 deletion on LEC in their experimental setting? Was proliferation affected, as previously reported?

- Additional investigation of key functional events regulated by CXCL11/12-ACKR3 axis would clarify, for example, whether leukocyte migration is affected by ACKR3-deletion.

2. Authors showed that primary LECs isolated from ACKR3i�LEC mice are equipped with AM1 receptor. Is the uptake of adrenomedullin affected in primary LEC from ACKR3i�LEC compared to control mice?

3. Typo in the abstract: “Specifically, ACKR-3-deficency”.

Reviewer #2: Sigmund et al here investigate the expression pattern and function of ACKR3, which is a scavenging receptor for chemokine and non-chemokine ligands, in lymphatics. They show that ACKR3 is widely expressed in mature lymphatics. Surprisingly, their data suggest that ACKR3 in LECs does not contribute to postnatal lymphangiogenesis and lymphatic drainage function. Overall the manuscript is well constructed and written. There are several areas that could benefit from additional experiments or clarification to improve the quality of the study.

Major issues:

1) Fig. 1B: lymphatic precollectors also express LYVE-1 although the expression is decreased compared to lymphatic capillaries (Lutter et. al J. Cell Biol., 2012). Please clarify.

2) Fig. 3A-D: Cre-driven recombination expression of RFP does not necessarily correspond to deletion of ACKR3 as the two pairs of loxP sites in Fig. 3A are independent to each other. The authors should validate ACKR3 deletion efficiency in the KOs by testing ACKR3 mRNA expression levels in the isolated dermal LECs.

3) Fig. S5: Postnatal lymphangiogenesis in diaphragm is pronounced from P5 to P7 (Ochsenbein et al Angiogenesis, 2016). Therefore, analysis of diaphragmic lymphatics from mice older than P5 (e.g. P7) is suggested. Was the age of Fig. S6 mice also P5? If yes, this issue similarly affects mesenteric lymphatics.

4) Page 9, statistical data analysis: In this study, normal distribution of the presented data cannot be assumed because the n values are low and the Gaussian distribution analysis is not applicable. Therefore Student’s t-test analysis is not valid here. Instead, the mann-whitney U-test that does not require assumption of normal distribution of the data is appropriate. In Fig. 4C, pairing delta MFI data from WTs and KOs is not valid, either. Please correct.

5) Pages 17-19, discussion: Differential expression pattern of ACKR3 in various lymphatic vasculature beds (dermal lymphatics v.s. lacteals, mesenteric and diaphragmic lymphatics, Figs. 2, S1-S3) has not been sufficiently discussed in the manuscript.

Minor issues:

1) Fig. 1 legends: “(D) ACKR3” should read “(C) ACKR3”. Please correct E-I accordingly.

2) There are multiple discrepancies between data shown in Fig.2 and the corresponding text. For example: It is unclear whether those GFP+ cells in the surrounding tissue can be identified as stromal cells; GFP signal is NOT ubiquitously present in CD31+ podoplanin- BECs (Fig.2A-E); Prox1 staining data is present in Fig. 2E instead of Fig. 2A-C. Please reconcile.

3) Page 6, line 12: “dermal skin cells” should read “dermal LECs”.

4) Page 11, line 12: “(Fig S2A and B)” should read “(Fig S2 and S3)”.

5) Page 13, line 6: “dermal cells” should read “dermal LECs”.

6) Page 14, line 18: “(Fig S5A)” should read “(Fig. S5C)”.

7) Fig. S4A. “stomal” should read “stromal”.

6. PLOS authors have the option to publish the peer review history of their article (what does this mean?). If published, this will include your full peer review and any attached files.

Reviewer #1: No

Reviewer #2: No

---

## [Author Response · Author response to Decision Letter 0]

26 Feb 2021

Response regarding additional requirements from PLOS One: 

1: Please ensure that your manuscript meets PLOS ONE's style requirements, including those for file naming. The PLOS ONE style templates can be found at

Response #1: We have updated the file names to fit PLOS ONE's style requirements.

2. To comply with PLOS ONE submissions requirements, in your Methods section, please provide additional information regarding the experiments involving animals and ensure you have included details on (1) methods of sacrifice and (2) the number of animals used in the experiment.

Additionally, please consider modifying your title to ensure that it is specific, descriptive, and concise (for example by specifying the animal model used)

Response #2: We have included information regarding method of sacrifice (p.5, top) and numbers of animals used in the experiments in the Figure legends 

Additionally, we have included the words “in mice” in the title, to be more specific and concise. The title now reads: “Lymphatic endothelial-cell expressed ACKR3 is dispensable for postnatal lymphangiogenesis and lymphatic drainage function in mice” 

Response #3: The data presented in this study will be made openly available in the ETH Research Collection (https://www.research-collection.ethz.ch/) upon acceptance. At this point, we will be able to generate and receive a DOI. 

Response #4: We included the “data not shown”, by showing the average weight of ACKR3i∆LEC compared ACKR3WT animals in S4 E Fig. (mentioned on p. 13, top). 

Response #5: We have included the captions for our supporting Information files at the end of our manuscript and matched the corresponding in-text citations. 

Comments from the Reviewers: 

Response to comments from Reviewer 1:

Reviewer #1: Sigmund et al. investigated the expression and function of the atypical chemokine receptor ACKR3 in the lymphatic vasculature of adult mice. The authors show that ACKR3 is widely expressed in mature lymphatics and that it exerts chemokine-scavenging activity in cultured murine skin-derived LECs. They generated and validated a lymphatic-specific, inducible ACKR3 knockout mouse that revealed no contribution of LEC-expressed ACKR3 to postnatal lymphangiogenesis, lymphatic morphology and drainage function.

This is the first study reporting ACKR3 expression and function in postnatal lymphatic vessels. Experiments and statistics are described in sufficient detail. Conclusions are presented in an appropriate fashion and are supported by the data. Limitations of the study are also discussed. However, the study would benefit from clarifying a few points:

1.1.) Authors report that CXCL11/12-AF647 uptake is abrogated in primary LECs from tamoxifen-induced ACKR3i∆LEC mice, demonstrating that the tamoxifen was effective in inducing ACKR3-deletion. - What is the functional impact of ACKR3 deletion on LEC in their experimental setting? Was proliferation affected, as previously reported?

Response #1.1: Overall, our data suggest that ACKR3 deletion in LECs does not impact LEC proliferation during postnatal lymphangiogenesis or LEC proliferation in LVs in the adult ear skin, as we did not observe LV hyperplasia or abnormal growth of vessels that would indicate such a phenotype. To more specifically address the Reviewer’s question, we have now additionally performed in vitro proliferation experiments to directly investigate the impact of ACKR3-deficiency on LEC proliferation.

Proliferation was assessed by Ki67 staining performed on in vitro-cultured primary LECs isolated from the tail skin of ACKR3i∆LEC and WT littermates. The new data clearly show that in vitro proliferation is unaffected in ACKR3i∆LEC, in support of the previous findings of our study. The new data are now shown in Figure S4 Fig F-G and mentioned on page 13 (top). 

1.2.) Additional investigation of key functional events regulated by CXCL11/12-ACKR3 axis would clarify, for example, whether leukocyte migration is affected by ACKR3-deletion.

Response #1.2: Sparked by the Reviewer’s comments we have investigated whether dendritic cell (DC) migration is affected by LEC-specific KO of ACKR3. To this end, we performed adoptive transfer experiments with bone marrow-derived DCs. Prior to those experiments, we first confirmed that the ACKR3-ligand CXCL12, which reportedly is also expressed by afferent lymphatics and mediates DC migration to draining lymph nodes (Kabashima et al., AJP 2017) can mediate transmigration of BM-DCs through a monolayer of immortalized murine LECs in vitro. However, when studying the migration of adoptively transferred DCs from the footpad to the draining popliteal LN, we found no difference in the number of migratory DCs that arrived in the LN in ACKR3i∆LEC compared to ACKR3WT mice (neither in steady-state nor during TPA-induced inflammation of the footpath). Thus, in the experimental model analyzed, no role for ACKR3 in DC migration through afferent lymphatics could be detected. 

We have now added these data as new Supplemental Figure S7 and are briefly mentioning the results in the discussion (page 21, top). 

2) Authors showed that primary LECs isolated from ACKR3i∆LEC mice are equipped with AM1 receptor. Is the uptake of adrenomedullin affected in primary LEC from ACKR3i∆LEC compared to control mice?

Response#2: Unfortunately, in spite of several attempts that also involved collaborators with profound expertise in protein labelling (i.e. Marcus Thelen, one of our co-authors), we did not manage to label recombinant AM in a way that it would retain its functionality after labeling (i.e. it lost its ability to induce LEC proliferation, presumably because the protein aggregated after the labeling). Thus, we were unfortunately not able to perform any AM uptake studies and to address this question of the Reviewer. 

However, considering that our data showed that uptake of the chimeric chemokine CXCL11/12 was lost in ACKR3i∆LEC(Fig 4A-C), together with the new data showing loss of ackr3 mRNA in ACKR3i∆LEC (S4 Fig C-D), we think that there now is sufficient evidence showing that ACKR3 was indeed largely lost in LECs upon tamoxifen treatment. 

Comment #3: 

3) Typo in the abstract: “Specifically, ACKR-3-deficency”

We thank the Reviewer for noticing this mistake! It has been corrected in the revised abstract to read “ACKR3-deficiency”.

 

Response to comments of Reviewer 2:

Reviewer #2: 

Sigmund et al here investigate the expression pattern and function of ACKR3, which is a scavenging receptor for chemokine and non-chemokine ligands, in lymphatics. They show that ACKR3 is widely expressed in mature lymphatics. Surprisingly, their data suggest that ACKR3 in LECs does not contribute to postnatal lymphangiogenesis and lymphatic drainage function. Overall the manuscript is well constructed and written. There are several areas that could benefit from additional experiments or clarification to improve the quality of the study.

Major issues raised by reviewer 2

1) Fig. 1B: lymphatic precollectors also express LYVE-1 although the expression is decreased compared to lymphatic capillaries (Lutter et. al J. Cell Biol., 2012). Please clarify.

Response #1: Indeed, pre-collectors have been found to express intermediate (lower) levels of LYVE-1 (Lutter, et al. J. Cell Biol., 2012). However, for our FACS-sorting (Fig. 1b), we only sorted the LYVE-1-negative and LYVE-1-positive podoplanin+CD31+ LECs, leaving out the population of LYVE-1-intermediate cells, which presumably make up the pre-collector LECs 

We are now specifically mentioning this in the results section and providing the Reference for the pre-collector LECs (p.10, top). 

2) Fig. 3A-D: Cre-driven recombination expression of RFP does not necessarily correspond to deletion of ACKR3 as the two pairs of loxP sites in Fig. 3A are independent to each other. The authors should validate ACKR3 deletion efficiency in the KOs by testing ACKR3 mRNA expression levels in the isolated dermal LECs.

Response #2: In addition to the data in Figure 4B and C showing reduced CXCL11/12 chemokine uptake in in vitro-cultured dermal LECs of ACKR3i∆LEC mice, which is a measure of reduced / abrogated ACKR3 protein expression in LECs of ACKR3i∆LEC mice, we now also included qPCR data showing reduced ACKR3 mRNA expression in isolated tail and lymph node (LN) LECs of ACKR3i∆LEC mice compared to WT animals. The new data are presented in S4 Fig C-D and reported in the results section (p.13, top). 

3) Fig. S5: Postnatal lymphangiogenesis in diaphragm is pronounced from P5 to P7 (Ochsenbein et al Angiogenesis, 2016). Therefore, analysis of diaphragmic lymphatics from mice older than P5 (e.g. P7) is suggested. Was the age of Fig. S6 mice also P5? If yes, this issue similarly affects mesenteric lymphatics.

Response #3: When we designed the experiment and chose the time point for the analysis of postnatal lymphangiogenesis, we specifically chose p5 as an optimal time for analysis because, in our hands, the developing vessel network in the diaphragm already starts to close at p5 and was considerably ramified and total area /size-wise further developed than what was reported / depicted in Ochsenbein et al. (2016) (Figures 1 and Figure 2). Notably, we have already successfully used this model, including the analysis at the p5 time point, in a recent publication from our group (Willrodt, A. H. et al. Front Immunol (2019), in which also Alexandra Ochsenbein was a co-author, as she taught us the technique. 

In our opinion / our hands, the p5 time point was best suited to detect possible delays or maladaptive lymphangiogenesis. Regarding the mesentery lymphatic development, formation of mesentery lymphatic vessels starts earlier than in the diaphragm, i.e. already at E14.5. In WT animals the mesenteric lymphatics have developed into a hierarchical, complex lymphatic network by E.18.5 (see for example Norrmén et al., 2009 J Cell Biol (2009) or Wang et al. Development (2016)). Thus, at p5, the main developmental process of vessel formation, involving proliferation of LECs, has already been completed - although the mesenteric lymphatic vessels still mature and valve formation is still continuing at this time point (Sabine et al. 2015). Thus, in the case of the mesenteric lymphatic vessels we believe, that we covered the time after primary vessel formation/ proliferation.

4) Page 9, statistical data analysis: In this study, normal distribution of the presented data cannot be assumed because the n values are low and the Gaussian distribution analysis is not applicable. Therefore, Student’s t-test analysis is not valid here. Instead, the Mann Whitney U-test that does not require assumption of normal distribution of the data is appropriate. In Fig. 4C, pairing delta MFI data from WTs and KOs is not valid, either. Please correct.

Response #4: We thank the Reviewer for this comment. We have reconsidered the use of all statistical tests and have revised our statistics paragraph accordingly (page 9, bottom). Importantly, we only apply Student`s test on normally distributed data with sufficiently high n values. The use of a particular statistical test is now also reported in each Figure legend. 

Regarding Figure 4, we have now omitted the pairing in Figure 4C and performed a Mann-Whitney U-test, as recommended by the Reviewer. In the case of Figure 4B, we insist that these data are paired, since the MFI baseline of the flow cytometric analysis can significantly vary between experiments, depending on the cytometer calibration and the batch of fluorescent chemokine used. Even more so, the primary cells used, were always derived from the same isolation/ culture. We have therefore applied a Wilcoxon matched pairs test, taking into consideration that the n value is low and the data are hence not normally distributed. 

5) Pages 17-19, discussion: Differential expression pattern of ACKR3 in various lymphatic vasculature beds (dermal lymphatics v.s. lacteals, mesenteric and diaphragmic lymphatics, Figs. 2, S1-S3) has not been sufficiently discussed in the manuscript.

Response #5: We have now inserted a new paragraph addressing this in the discussion (page 18, bottom). 

Minor issues raised by Reviewer 2:

1) Fig. 1 legends: “(D) ACKR3” should read “(C) ACKR3”. Please correct E-I accordingly.

Thanks for noticing. This issue has now been corrected in the Figure legend of the revised manuscript.

2) There are multiple discrepancies between data shown in Fig.2 and the corresponding text. For example: It is unclear whether those GFP+ cells in the surrounding tissue can be identified as stromal cells; GFP signal is NOT ubiquitously present in CD31+ podoplanin- BECs (Fig.2A-E); Prox1 staining data is present in Fig. 2E instead of Fig. 2A-C. Please reconcile.

We believe this mistake happened because we exchanged some of the images during our internal revision, and we are very thankful to the Reviewer for pointing out that the corresponding text does not match some of the images anymore. To address the point raised, namely that GFP was not clearly visible in CD31+ Podoplanin- BECs in the representative images we selected, we have now added an additional image panel (Fig 2D), that clearly demonstrates GFP expression in some (but not all) CD31+ blood vessels. 

The text corresponding to Figure 2 was revised as follows (p. 11, middle):

“Notably, GFP was not exclusively expressed by LECs in ACKR3 GFP/+ reporter mice, but was frequently also found in unidentified or stromal cells (Fig 2A, C-D) in the surrounding tissue and associated with nerves positive for Tuj1 (Fig 2C, D). Moreover, GFP was also detected in many but not all podoplanin- / CD31+ blood vessels (Fig 2B, C) and in the epidermal layer of the skin, indicating that ACKR3 is expressed by keratinocytes (Fig 2E). GFP expression was additionally detected in large lymphatic collectors such as the flank collectors of ACKR3 GFP/+ reporter mice (Fig 2F).”

3) Page 6, line 12: “dermal skin cells” should read “dermal LECs”.

In the experiment described, we performed the chemokine uptake with a mixture of isolated dermal stromal cells, still containing BECs and other stromal cells such as fibroblasts. LECs were identified by flow cytometry using anti-CD31 and anti-podoplanin staining. The reason, why we did not perform this assay with purified LECs is because this allowed us to test uptake of fluorescent CXCL11/12 chemokine at earlier passage numbers (i.e. sooner after isolation), what was important, since we observed that upon culture LECs gradually lose ACKR3 expression. 

4) Page 11, line 12: “(Fig S2A and B)” should read “(Fig S2 and S3)”.

Thank you. It has been corrected in the revised manuscript.

5) Page 13, line 6: “dermal cells” should read “dermal LECs”.

Same as point 3. In this experiment we isolated a dermal stromal cell mixture and subsequently identified cell types (including LECs) by antibody staining and flow cytometry.

6) Page 14, line 18: “(Fig S5A)” should read “(Fig. S5C)”.

Thank you. It has been corrected in the revised manuscript and Figure: we have changed Figure panels to fit the text flow in the manuscript.

7) Fig. S4A. “stomal” should read “stromal”.

Thanks for noticing. It has been corrected in the revised Figure legend.

---

## [Decision Letter · Decision Letter 1]

11 Mar 2021

Lymphatic endothelial-cell expressed ACKR3 is dispensable for postnatal lymphangiogenesis and lymphatic drainage function in mice

PONE-D-20-24551R1

Dear Dr. Halin,

We’re pleased to inform you that your manuscript has been judged scientifically suitable for publication and will be formally accepted for publication once it meets all outstanding technical requirements.

Kind regards,

Jean-Léon Thomas

Academic Editor

PLOS ONE

Additional Editor Comments (optional):

All the comments of the two reviewers have been addressed in the revised manuscript. Congratulations to the authors for their excellent work.

Reviewers' comments:

Reviewer's Responses to Questions

**Comments to the Author**

1. If the authors have adequately addressed your comments raised in a previous round of review and you feel that this manuscript is now acceptable for publication, you may indicate that here to bypass the “Comments to the Author” section, enter your conflict of interest statement in the “Confidential to Editor” section, and submit your "Accept" recommendation.

Reviewer #1: All comments have been addressed

Reviewer #2: All comments have been addressed

2. Is the manuscript technically sound, and do the data support the conclusions?

Reviewer #1: Yes

Reviewer #2: Yes

3. Has the statistical analysis been performed appropriately and rigorously? 

Reviewer #1: Yes

Reviewer #2: Yes

4. Have the authors made all data underlying the findings in their manuscript fully available?

Reviewer #1: Yes

Reviewer #2: Yes

5. Is the manuscript presented in an intelligible fashion and written in standard English?

Reviewer #1: Yes

Reviewer #2: Yes

6. Review Comments to the Author

Reviewer #1: Authors have adequately addressed all comments. The manuscript advances our understanding of regulation of lymphatic endothelial cells. The data presented in the manuscript supports authors conclusions. Experiments have been conducted rigorously, with appropriate controls, replication, and sample sizes. The conclusions are appropriately based on the data presented. Manuscript is presented in an intelligible fashion and written English.

Reviewer #2: The authors have adequately addressed all my concerns. I think this manuscript is now suitable for publication.

7. PLOS authors have the option to publish the peer review history of their article (what does this mean?). If published, this will include your full peer review and any attached files.

Reviewer #1: No

Reviewer #2: No

---

## [Editor Report · Acceptance letter]

6 Apr 2021

PONE-D-20-24551R1 

Lymphatic endothelial-cell expressed ACKR3 is dispensable for postnatal lymphangiogenesis and lymphatic drainage function in mice 

Dear Dr. Halin:

I'm pleased to inform you that your manuscript has been deemed suitable for publication in PLOS ONE. Congratulations! Your manuscript is now with our production department. 

Kind regards, 

on behalf of

Dr. Jean-Léon Thomas 

Academic Editor

PLOS ONE